# Laser Cutting Coupled with Electro-Exfoliation to Prepare Versatile Planar Graphene Electrodes for Energy Storage

**DOI:** 10.3390/ijms24065599

**Published:** 2023-03-15

**Authors:** Jianren Wang, Tianshuo Yang, Neus Vilà, Alain Walcarius

**Affiliations:** 1Université de Lorraine, CNRS, Laboratoire de Chimie Physique et Microbiologie pour les Matériaux et l’Environnement (LCPME), F-54000 Nancy, France; wangjr@ysu.edu.cn (J.W.); neus.vila@univ-lorraine.fr (N.V.); 2Key Laboratory of Applied Chemistry, School of Environmental and Chemical Engineering, Yanshan University, Qinhuangdao 066004, China

**Keywords:** pseudocapacitive materials, graphene, planar electrode

## Abstract

The study of planar energy storage devices, characterized by low-cost, high capacity, and satisfactory flexibility, is becoming a valuable research hotspot. Graphene, monolayer sp^2^ hybrid carbon atoms with a large surface area, always acts as its active component, yet there is a tension between its high conductivity and ease of implementation. Although the difficult-to-assemble graphene can easily achieve planar assemblies in its highly oxidized form (GO), the undesirable conductivity, even after proper reduction, still restricts its further applications. Here, a facile “Top-down” method has been proposed to prepare the graphene planar electrode via in situ electro-exfoliation of graphite supported on a piece of laser-cutting patterned scotch tape. Detailed characterizations have been performed to study its physiochemical property evolution during electro-exfoliation. The obtained flexible graphene planar electrodes show decent energy storage performance, e.g., 40.8 mF cm^−2^ at a current density of 0.5 mA cm^−2^ and an 81% capacity retention at a current density of 8 mA cm^−2^ for the optimized sample G-240. Their high conductivity also makes it possible to couple them with other redox-active materials through electrodeposition to improve their performance, e.g., ferrocene-functionalized mesoporous silica film (Fc-MS), MnO_2_, and polyaniline (PANI). The highest capacity was achieved with the PANI functionalized sample, which achieved a 22-fold capacity increase. In a word, the versatility, practicality, and adaptability of the protocol to prepare the planar graphene electrode proposed in this work make it a potential candidate to meet the continuously growing energy storage demands.

## 1. Introduction

The last few decades have witnessed the continuously surging demands of flexible and wearable microelectronics in various fields, such as biomedical sensors, wearable products, sports assist devices, etc. [1,2]. Therefore, beyond higher energy and power densities, these cutting-edge applications put forward more rigorous requirements for energy storage systems, including ease of miniaturization, robust mechanical properties, and high flexibility [3]. Generally, conventional energy storage devices, mostly composed of vertical sandwich structures with two electrodes and one separator membrane and filled with liquid electrolyte, are too difficult to be fabricated into flexible micro-devices due to the complexity of such configuration [4]. In contrast, in-plane interdigitated devices, comprising flatly aligned but spatially separated energy storage materials with an electrolyte coating directly on the side, have gained more and more research interest in recent years. This idiosyncratic design offers several merits over the traditional sandwich structure in terms of emerging application scenarios, such as:
the distance between the adjacent working and counter electrodes could be minimized to some extent through advanced patterning techniques, which could reduce the internal resistance of the devices and improve their power output [5];the monolayer nature of the in-plane structure could largely increase the volume energy density by decreasing the thickness of devices and also reduce short-circuit risks triggered by external forces;the simplified architecture also makes it easier for incorporation into electronic devices and even makes it suitable as a wearable power source when utilizing superiorly flexible supporting materials as substrates that can bear the long-term twisting and folding distortions.

Featured with a large surface area (~2630 m^2^ g^−1^), excellent electrical conductivity (~104 S cm^−1^), and robust mechanical properties (Young’s modulus of ~1TP) [6], graphene is widely used as the active material for flexible in-plane energy storage devices when integrated on various flexible substrates, such as polyethylene terephthalate (PET), polydimethylsiloxane (PDMS), and textiles, for instance [7,8,9]. Instead of directly utilizing graphene, graphene oxide is mostly used as the starting material to assemble these devices owing to its abundant functional groups, hydrophilic property, and high surface adhesion energy [10]. However, after the reduction processes, the products, so-called reduced graphene oxide, cannot recover the pristine graphene structure either in thermal, chemical, or electrochemical processes; therefore, they always suffer from relatively low conductivities [11]. In addition, one more technical difficulty in fabricating such devices lies in precisely patterning graphene on substrates to make full use of the surface space while avoiding the short-circuit problem. Several possible strategies have been developed in recent years, including chemically coating graphene on pre-patterned current collectors [12], laser writing on in situ reducing graphene oxide [13], plasma etching [14], and inkjet printing [15]. Nevertheless, these methods still face some drawbacks: the dependence on sophisticated facilities, the complicated preparation processes, the utilization of organic solvents, and the limited graphene-loading amount. As such, developing a low-cost and facile method to assemble high-quality graphene on flexible substrates in a dense and patterned way is pivotal to promoting the development of this kind of device.

Electro-exfoliation is an efficient protocol to produce various types of two-dimensional materials from their bulk counterparts in a scalable fashion [16]. This method can also be used to exfoliate bulk graphite into mono-/few-layer structures through a coupling of electric field-driven sulfate ion intercalation and anodic water oxidation processes [17]. The flakes produced with such a method always show a high C/O ratio (~20) over other “Top-down” chemical methods (e.g., ~2 of Hummers’ method) [18], indicative of a small number of defects. As reported previously, only the edges of the flakes can be oxidized [19]; therefore, their properties, particularly electrical conductivity, should be more similar to graphene than graphene oxide [19], making them good candidates for energy storage electrodes. However, the as-formed graphene flakes are more likely to peel in the solution phase due to the intense bubbles produced by the constant water decomposition and the lack of binding sites at the graphite electrode. Although an “in-situ exfoliate and deposit” method has been developed to assemble vertically aligned graphene on a conductive substrate [20], the graphene density of the final electrode is still relatively low, leading to a limited areal capacitance. If one tears a piece of graphite film with scotch tape to form a composite, the sticky tape surface would prevent the flakes from stripping during the electro-exfoliation process, creating a scotch tape-confined densely packed graphene electrode; this is what we plan to investigate in the present work. Moreover, the underlying scotch substrate also has several advantages, as discussed hereafter. The inherent characteristics of scotch tape permit this electrode a high degree of flexibility and a robust mechanical nature, which makes it suitable for increasingly complex application scenarios. The pattern of the scotch tape can even be finely edited with a laser-cutting technique, which can be a low-cost, time-saving, and versatile way to fabricate the in-plane planar devices. The high conductivity and large surface area of the graphene layer also make it easy to further composite with other energy-storage active materials. As a proof of concept, we will evaluate here some modifiers that can be easily electrodeposited, such as manganese dioxide (MnO_2_), polyaniline (PANI), and redox molecules (i.e., ferrocene moieties)-functionalized mesoporous silica films. Overall, a complete protocol to fabricate the planar graphene and/or its composite energy storage electrodes is proposed here, and the obtained electrodes not only show decent energy storage abilities but also permit adequate mechanical properties, which may have the opportunity to be used in next-generation flatly flexible energy storage devices.

## 2. Results and Discussion

### 2.1. Preparation of the Planar Graphite Electrodes

The electrode preparation process is illustrated in Figure 1a, and as one can see, the laser first easily engraves the scotch tape to obtain different design patterns, which can be further used to peel off a piece of graphite film from graphite foil via its sticky surface. The resolution of this technique, in this study, could reach 800 μm, which suits the miniaturization of next-generation energy storage devices. Please also note that there is still room for optimization, and a higher resolution might be achieved with the advanced laser generator.

This graphite–scotch tape composite undergoes the following electro-exfoliation process to fabricate the final flatly flexible graphene electrodes. Figure 1b gives the current profile of the electro-exfoliation process along with the corresponding digital photos of the electrode at a certain time. One can see that after a fast drop from 75 mA cm^−2^ to 60 mA cm^−2^ within a few seconds, the exfoliation current gradually goes down from 60 mA cm^−2^ to 10 mA cm^−2^ until the exfoliation time reaches ca. 250 s and then reaches a steady state after that. The initial sharp fall of the current is probably due to the formation of an electrical double-layer process, which could normally finish the charging within a few seconds. Afterward, the anodic current mainly comes from the sulfate ions inserted into the graphite interlayers and the oxygen evolution reaction [21], both of which help to expand and separate the graphite layer into mono-/few-layer structures of graphene. The continuously decreasing current indicates the consumption of the graphite reactant and the increasing resistance of the graphite/scotch electrode because the inserted sulfate ions inside the exfoliated graphene could deteriorate the conductivity, as confirmed via the later activation experiments. The following steady state of the current could be a signal that all the graphite on the scotch tape has been converted into graphene. The residue current could be a balance of the inserted sulfate ions between the electrical driving force and the thermodynamic distribution. Different from the almost constant current of directly electro-exfoliating graphite foil (Appendix A), the gradually decreasing current and the almost clear final solution (Appendix A) indicate the scotch tape confinement effect towards its onside graphite, and the following characterizations also prove this. The digital photos of the electrode with distinct exfoliation times show the color of the exfoliated material: the part above the yellow line gets darker with prolonged exfoliation, indicating a structural change. 

Taking the 240 s exfoliation sample (G-240) as an example, the conductivity change was studied using both the AC impedance (Figure 1c) and the DC I–V curve (Figure 1d) to prove the need for the activation process at the potential of 0V (vs. Ag/AgCl, 3 M). From the Nyquist plot of the impedance spectra, two kinds of shape can be observed, and an obvious semicircle can be observed for the G-240 before activation of the sample, which may come from the charge transfer between graphite and the intercalated sulfate ions. Therefore, two equivalent circuits were used to fit them. An extra R_ct_ element in parallel to the constant phase element (CPE) was added in the equivalent circuit for the graphite and the G-240 sample. Their corresponding circuits and detailed fitting results are given in Appendix A. As can be seen, the resistance of the graphite/scotch electrode is ~0.7 Ω, while its resistance surges to as high as 1145 Ω after the electro-exfoliation process, which might be due to deterioration from the intercalated sulfate ions. Fortunately, the intercalated sulfate ions can be reversibly repelled out of the solution by applying a 0 V potential on the side for another 240 s; the current of this activation process is shown in Appendix A. A strong cathodic current density ranging from −35 mA cm^−2^ to −1 mA cm^−2^ can be first observed at 40 s, which may come from the fast move-out of the negatively charged sulfate ions located at the surface/near-surface of the graphene. For the period from 40 s to 240 s, the current delivers a slowly decreasing trend from −1 mA cm^−2^ to −0.25 mA cm^−2^, and the much smaller but relatively steady current may reflect the leaching of sulfate ions from the lattice of a few layers of graphene and un-exfoliated graphite. After the activation, most sulfate ions are removed from the graphene electrode, and the conductivity of G-240 recovers back to 2.21 Ω. The similar resistance change in this process can also be confirmed via the DC I-V test, where two platinum clamps with 0.5 cm in width effectively contact the targeted sample at a distance of 2 cm. The resistances of the graphite electrode, G-240, and G-240 before activation are, respectively, 3.1 Ω, 8.1 Ω, and 20,740.8 Ω in our test conditions. 

In addition, the relationship between the exfoliation time and the electroactive area was analyzed using the cyclic voltammetry (CV) method (Figure 1e). By prolonging the exfoliation time, the corresponding sample shows that the current response steadily rises to an 11-fold increase until 240 s and then reaches a stable state afterward. (Please note that the active area here is not the actual area of the electrode but only reflects the increased times of the area.) The exfoliation current shows an opposite trend: it decreases first and then stays almost stable. This phenomenon not only proves the feasibility of using electro-exfoliation to prepare large surface-area graphene electrodes but also demonstrates that the exfoliation current could be used as an indicator to evaluate the density of graphene. The G-240 is therefore the optimized sample, and its related physiochemical properties as well as energy storage abilities are further evaluated in the following section.

### 2.2. Physico-Chemical Characterization of the Planar Graphite Electrodes

The proposed graphene formation process during the electro-exfoliation is illustrated in Figure 2a. The intercalation of sulfate ions inside the interlayer of graphite would weaken the interaction between the adjacent graphene layers, and the simultaneous oxygen gas evolution would break them down into graphene. The XRD pattern variations with different exfoliation times (Appendix A) prove this assumption to some extent, as the decreased intensity and the widening half-peak width of the (002) peak with an extension of the exfoliation time indicate that the insertion of sulfate ions will affect the integrity of the graphite lattice. In addition, from our observation, long-time exfoliation will also lead to a stronger peak at ca. 17°, which might be due to the formation of the sulfate-intercalated graphite [22]. 

The microstructures of the graphite/scotch electrode and G-240 electrode were characterized through scanning electron microscopy (SEM). The cross-section view of the original graphite/scotch tape electrode (Appendix A) shows a compact graphite layer with a thickness of ~50 μm, and the underlying brightest region that arose from the charge accumulation effect is the insulating scotch tape substrate. After the exfoliation, the cross-section of G-240 (Figure 2b) shows that the material becomes heterogeneous. The substrate is composed of two distinct layers: the highly porous top layer, with a thickness that swelled to ca. 110 μm due to the oxygen bubbles’ impact, and the layer near the scotch tape, with a thickness of ca. 15 μm, which seemed not to be affected by the exfoliation process and continued to manifest the pristine graphite structure due to the scotch tape confinement effect. Further, the high-resolution SEM image (Figure 2c) of the top layer also reveals it is composed of a few layers of graphene, which is also confirmed by the AFM result (Figure 2d) with a thickness of 5~8 nm and the almost transparent flakes in the TEM image (Appendix A). The anisotropic structure feature of G-240 could benefit its energy storage application, notably, because the highly porous graphene top layer will not only possess continuous channels for counter ion diffusion but also provide a large surface area for the energy storage or even compositing with other active materials. In addition, Raman spectra (Figure 2e) demonstrate that electro-exfoliation will also alter the carbon hybridization status as the ratio of D-band (sp3 hybridization)/G-band (sp2 hybridization) increases with an extension of the exfoliation time. The gradually increasing D-band might arise from the surface oxidation of the carbon basal plane and/or the edge sites of the newly formed graphene [23]. The 2D band, sensitive to the layer of graphene, has also been observed in different samples. For monolayer graphene, its intensity should be much higher than that of the G band. However, this is not the case for our samples, and similarly low-intensity 2D bands have been observed for all samples. This can be ascribed to the composite nature of our samples, as the graphite layer next to the scotch tape cannot be exfoliated. Despite this, a noticeable shape change can be observed for different exfoliation times. The shoulder peak of graphite at the lower Raman shift (P_1_) gradually increases with an extending exfoliation time, indicative of the formation of few-layer-structured graphene [24], while the intensity of the P_2_ peak gradually decreases, suggesting the reduced ratio of graphite in the composites.

As mentioned above, G-240 is the most “cost-effective” sample, and therefore its energy storage properties were evaluated in detail. The CV curves of G-240 were recorded at various scan rates ranging from 5 mV s^−1^ to 200 mV s^−1^ in a 1M H_2_SO_4_ aqueous solution. The corresponding results (Figure 2f) show a rectangular shape, despite a pair of very small redox peaks at ca. 0.35 V coming from oxygen-containing functional groups [25] (to clarify this pair of peaks, the CV curve at a low scan rate of 1 mV/s is given in Appendix A), which occur at both the relatively low scan rate of 5 mV s^−1^ and at the high scan rate of 200 mV s^−1^. This behavior is a double-layer type process, where the charges accumulate on the surface of the conductive surface and the counter ions are physically adsorbed on the side of the surface of the electrode by the potential-driven force in the meantime. There is no charge that could transfer across the interface during the formation of the double layer, and therefore its behavior can be described as that of a capacitor in the equivalent circuit. In addition, the well-maintained rectangular shape and the nearly 90 °C turning at the switching potential (even 82 °C at the scan rate of 200 mV s^−1^) further demonstrate the negligible internal resistance and fast ion diffusion process during the charge–discharge processes. The energy storage ability of G-240 was assessed with the galvanostatic charge–discharge technique (Figure 2g) with current densities from 0.5 mA cm^−2^ to 8 mA cm^−2^. Despite a slight distortion at ca. 0.35 V, the symmetrically triangular charge-discharge curves show the pure electrical double-layer energy storage process and high coulombic efficiency. Based on these curves, G-240 shows a high-rate performance with 40.8 mF cm^−2^ at a current density of 0.5 mA cm^−2^ (49 s charge/discharge time) and 81% (33.0 mF cm^−2^) retention at a high current density of 8 mA cm^−2^ (2 s charge/discharge time). Unsurprisingly, G-240 delivered a robust cycling performance (Appendix A). In addition, there is no obvious difference between the impedance of the original G-240 and that after its long-term cycling, which indicates its stable structure (Appendix A).

### 2.3. Modification of the Planar Graphite Electrodes to Get Pseudocapacitive Materials

Rather than using them alone, as previously mentioned, the large surface area and good electrical conductivity of graphene electrodes make it possible to further turn them into nanocomposites by associating them with other redox-active materials in order to improve their energy storage performance. Facile electrodeposition methods were adopted for the modification of G-240, respectively, with ferrocene-functionalized mesoporous silica film (Fc-MS), manganese dioxide (MnO_2_), and polyaniline (PANI), as illustrated in Figure 3a. 

The corresponding SEM images (Figure 3b,d,f) and element mapping results (Appendix A) reveal that all three redox-active components were well integrated on the surface of graphene, yet in different forms. The Fc-MS and MnO_2_ cover the graphene nanosheets in the film with a thickness of ~70 nm (similar to our previous report [26]) in the former case and 130 nm in the latter case. In contrast, the electropolymerized PANI was found to independently grow on the graphene surface in a nanowire form. The loading masses of PANI and MnO_2_ on the side of G-240 are 1.9 mg/cm^2^ and 3.1 mg/cm^2^. Regarding the Fc-MS functionalized sample, the loading of the ferrocene molecules was a trace amount, which is below the detection limit of the balance. However, our previous work proved that almost all ferrocene molecules are redox-active [26]. Please note that, not limited to the three components given here, these results demonstrate that the prepared graphene electrode can act as a versatile substrate to manufacture composites with extensive nanomaterials. The energy storage performances of these three nanocomposites were further evaluated with chronopotentiometry, and the corresponding curves are given in Figure 3c,e,g. Different from that of the graphene electrodes, the shapes of all curves are distorted, no longer triangular shaped, due to the redox reactions at a certain potential, which will finally lead to a potential dependent capacitance. Although their shapes are still not the same as that of the battery-type electrodes, a coulomb represents a better unit to evaluate their performance, instead of a farad [25]. One can see, at any current densities, a much longer charge/discharge time can be obtained for the prepared composites than that of G-240, which means that much more charge can be stored inside (Figure 3h,i). The maximal capacity increases to ca. 22 times from 24.5 mC cm^−2^ for G-240 to 565.5 mC cm^−2^ for PANI-G at the current density of 0.5 mA cm^−2^. Certainly, different composites show different energy storage behaviors, and better performance could be achieved with further optimizations. All samples show more than 85% capacity retention after 1000 consecutive cycles at a scan rate of 100 mV s^−1^ (Appendix A). The impedance data of MnO_2_ and PANI samples before and after long-term cycling have been enriched in Appendix A. Their spectra are composed of a semicircle and a long tail, indicative of charge transfer and capacitor-like behavior. In addition, there is no obvious change between a pristine sample and that after cycling, which indicates their stable structure. Concerning the impedance of the Fc-MS functionalized sample, one can refer to our previous work [26,27].

The energy storage of the three composites involves charge transfer processes across the electrode interfaces, which is much more complicated than the pristine G-240 material. Therefore, the identification of the rate-determining step should be meaningful to improve their energy storage characteristics. The CV curves of the composites can maintain their originally intrinsic shapes only in a relatively slow scan rate range from mV s^−1^ to 20 mV s^−1^ (Figure 4a,d,g). As is the nature of the R-C circuit, serious polarization and the resistance-controlled behavior will occur at “high scan rates”, which is due to the increased surface capacitance as well as the resistance from the solution, electrode materials, and charge transfer processes. The definition of “high scan rates” is, however, different for different samples (Appendix A). As shown, obvious distortion occurs at a scan rate of 50 mV s^−1^ for the PANI sample, but the Fc-MS sample can still maintain its original shape at a scan rate of 200 mV s^−1^, which means the Fc-MS could be used for power-type energy storage devices, as discussed in our previous work in detail, yet for non-flexible electrodes [26,27]. The rate-determining step of the energy storage process in the low scan rates domain has also been analyzed by fitting the CV curves with the following Equation (1):*i* (*V*) = k_1_*v* + k_2_*v*^0.5^
(1)
where k_1_*v* and k_2_*v*^0.5^ represent the surface-controlled current and diffusion-controlled current, respectively. The fitting results (Figure 4b,e,h) reveal that 57%, 77%, and 90% of the total current come from the surface-controlled current at the scan rate of 10 mV s^−1^ for PANI, MnO_2_, and Fc-MS, respectively. This means that the ion diffusion resistances are also different during the energy storage process and that the non-existing solid-state ions diffusion for the Fc-MS sample gives a much higher surface-controlled current ratio, even similar to that of the G-240 electrode at 95 % (Appendix A). Similar trends can also be seen by fitting the redox peak currents vs. potential scan rates with Equation (2): *i* = a*v*^*b*^
(2)
where the empirical *b* value is an indicator of the type of reaction, ranging from 0.5 to 1 (0.5 for a diffusion-controlled behavior, and 1 for a surface-controlled charge transfer). The largest *b* value of Fc-MS (~0.91) again demonstrates a major surface-controlled redox process, in contrast to the diffusion-controlled behavior of the PANI sample (*b* value of ~0.58).

### 2.4. Evaluation of a Symmetrical Device

Two pieces of PANI-functionalized graphene electrodes were also assembled into a symmetrical device to evaluate the possible practical applications of the flexible graphene-based nanocomposite electrodes (Figure 5a–d). The same acidic electrolyte as that above in a three-electrode configuration was adopted to run the test on the assembled device. Figure 5a shows a group of CV curves from a scan rate of 5 mV s^−1^ to 200 mV s^−1^ in a potential window of −0.2 V to 0.9 V. A pair of broad redox peaks appear at ca. 0.3 V at the scan rate of 5 mV s^−1^, which is consistent with that observed with the three-electrode system (Figure 4g), although characterized by a somewhat larger anodic-to-cathodic peak-to-peak separation. Indeed, the peak polarization caused by the resistance is observed, especially when increasing the scan rate, but the rather symmetrical curves are still noticeable, even at the high scan rate of 200 mV s^−1^, indicative of an expected rate performance. The areal capacitance of the device is examined using chronopotentiometry by applying various current densities ranging from 0.5 mA cm^−2^ to 4 mA cm^−2^ in the same potential window as the CV tests, as plotted in Figure 5b. There is no obvious plateau, despite a little distortion, even at a small current density of mA cm^−2^, which demonstrates the pseudocapacitive nature of the device. Its derived capacitances are given in Figure 5c, and one can see a high areal capacitance of 79.8 mF cm^−2^ at a current density of 0.5 mA cm^−2^ and a 51.3% capacity retention at a current density of 4 mA cm^−2^. Its Ragone plot is given in Figure 5d for its easy comparison with other similar film-type devices [28,29,30,31,32]. The present one can deliver a robust cycling performance, reaching a capacity retention of 78% after 1000 consecutive cycles at a scan rate of 100 mV s^−1^ (Appendix A). Following this conceptual demonstration, future works could be directed toward replacing this ‘classically’ deposited PANI with PANI nanowire arrays exhibiting ultra-fast electrochemical responses [33].

## 3. Materials and Methods

### 3.1. Reagents

Tetraethoxysilane (TEOS, 98%), 3-chloropropyltrimethoxysilane (Cl-PTES, 97%), and graphite were purchased from Alfa Aesar (China), and cetyltrimethylammonium bromide (CTAB, 99%) was purchased from Acros (Japan). Acetonitrile (ACN, 99%), dimethylformamide (DMF, 99%), and cyclohexane (99%) were obtained from Merck. (American) Ethynylferrocene (97%), lithium perchlorate (95%), lithium chloride (95%), copper sulfate (99%), sodium diethyldithiocarbamate trihydrate, and HCl (37%) were purchased from Sigma-Aldrich. Sodium nitrate (99%), ascorbic acid (97%), tetrabutylammonium bromide (99%), and poly(vinyl alcohol) (PVA, MW = 72,000 g/mol) originated from Fluka. Graphite foils and scotch tape were purchased from Shanghai Carbon Co. Ltd (Shanghai, China). and 3M, respectively. Sodium sulfate anhydrous (99%), manganese sulfate (95%), concentrated sulfuric acid (98%), and aniline (99%) were from Meryer, Kaitong, Kermal, and Macklin, respectively. All compounds were used directly without further purification. The (3-azidopropyl) triethoxysilane (AzPTES) was synthesized with the protocol reported previously [34].

### 3.2. Apparatus

Transmission electron microscopy (TEM) investigation was performed with an ARM 200F Cold FEG TEM/STEM equipped with a GIF Quantum ER (JEOL, Japan). Scanning electron microscopy (SEM) analyses were carried out with a JSM-840 (JEOL, Japan) or SUPRA 55 apparatus equipped with an energy-dispersive X-ray (EDX) microanalyzer (Zeiss, Germany), and the high-resolution SEM micrographs were obtained with the model JSM-IT500HR apparatus (JEOL). Atomic force microscopy (AFM) analysis was performed with Asylum JEOL JSPM-5200. Raman spectroscopy measurements were carried out with a RenishawinVia spectrometer with a green light laser (532 nm). X-ray diffraction (XRD) pattern was obtained using the Nanoviewer from Rigaku (CuK α radiation) at a scanning rate of 5 °C/min.

### 3.3. Electrochemistry

Electrochemical experiments (including exfoliation of the graphite foil, electrodeposition of the silica film, PANI, MnO_2_, cyclic voltammetry, and charge-discharge measurements) were performed with Autolab PGSTAT-300, µAutolab, or Biologic SP-150 potentiostat. The Ag/AgCl (3.5 M KCl) electrode and platinum plate (1 cm × 1 cm) served as the reference electrode and counter electrode, respectively. An aqueous solution containing 1 M H_2_SO_4_ was used as the electrolyte for the performance test of the graphene samples and PANI/Fc-MS-functionalized samples, while 1 M Na_2_SO_4_ electrolyte was used for investigating the MnO_2_−functionalized sample.

### 3.4. Preparation of the Scotch Tape-Supported Graphene Samples (G-X, Where X Represents the Exfoliation Time)

The laser cutting technique was first used to pattern the scotch tape to a certain design. Subsequently, the as-patterned scotch tape was pasted on the surface graphite foil, followed by a peeling-off process to hand over a piece of graphite on the scotch tape. After that, similar to the electro-exfoliation method reported elsewhere [17], the electro-exfoliation process was further performed in 0.1 M Na_2_SO_4_ aqueous solution with a two-electrode system, where a piece of graphite-coated scotch tape (graphite/scotch electrode) and platinum mesh (2.5 cm × 2.5 cm) were used as the working and counter electrodes, respectively. The exfoliation was achieved by applying a voltage of 4 V for 240 s with a distance of 5 cm between the two electrodes. The obtained intermediate was highly resistive. To recover its conductivity, an activation process was used by applying 0 V to the intermediate for 240 s in a three-electrode system with Ag/AgCl as the reference to remove the inserted sulfate ions. Finally, the as-prepared electrode, labeled as G-240, was thoroughly washed with deionized water and dried at room temperature overnight. In addition, different exfoliation times of 30 s, 60 s, 120 s, and 360 s were also conducted to investigate the structural evolution, and the corresponding products were labeled as G-30, G-60, G-120, and G-360, respectively.

### 3.5. Preparation of the Graphene-Supported Ferrocene-Functionalized Silica Thin Film Electrode (Fc-MS-G)

The preparation protocol was similar to what we reported before [26], except a piece of G-240 was used as the working electrode in the present study. The final obtained composite was labeled as Fc-MS-G for simplicity.

### 3.6. Preparation of the Graphene-Supported MnO_2_ Electrode (MnO_2_-G)

Cyclic voltammetry was used to deposit MnO_2_ on the graphene surface with a three-electrode configuration, including a piece of G-240 as the working electrode, a platinum plate counter electrode, and Ag/AgCl (3 M) as the reference electrode. The deposition was carried out within a potential window ranging from 0 V to + 1.2 V at a scan rate of 10 mV s^−1^ in 0.1 M MnSO_4_. After 40 consecutive cycles and thorough cleaning with water, one can obtain the MnO_2_/graphene composite, labelled as MnO_2_-G for simplicity.

### 3.7. Preparation of the Graphene-Supported PANI Electrode (PANI-G)

The same configuration as the MnO_2_ preparation, but with a different electrolyte solution consisting of a mixture of 0.1 M H_2_SO_4_ and 0.01 M aniline, was used to prepare the graphene-supported PANI sample. The deposition process was achieved via 40 consecutive cycles in a potential window extending from −0.2 V to +1.0 V at a scan rate of 10 mV s^−1^. After thoroughly cleaning with water, one can obtain the PANI/graphene composite, labelled as PANI-G for simplicity.

## 4. Conclusions

In this work, a laser-cutting coupled with an electro-exfoliation protocol was proposed to prepare planar graphene-based energy storage electrodes with a resolution of 800 μm. Compared to the graphene electrodes reported elsewhere, the electrodes prepared with this “Top-down” method effectively solve the tension between high conductivity and easy processability. The optimized G-240 graphene sample shows a rate performance of 40.8 mF cm^−2^ at a current density of 0.5 mA cm^−2^ and 81% (33.0 mF cm^−2^) retention at a high current density of 8 mA cm^−2^. The capacity can be further improved via the formation of composites with other redox-active components, Fc-MS, MnO_2_, and PANI in this work; the maximum capacity can increase by 22 times to 565.5 mC cm^−2^ for the PANI-functionalized sample, for instance. Thanks to the flexibility of the scotch tape substrate, a flexible symmetric pseudo-capacitor with a pair of PANI-functionalized graphene electrodes was assembled, showing a capacitance of 79.8 mF cm^−2^. Such satisfactory results prove the protocol of this work in offering a new idea for the construction of flexible planar energy storage devices.

## Figures and Tables

**Figure 1 ijms-24-05599-f001:**
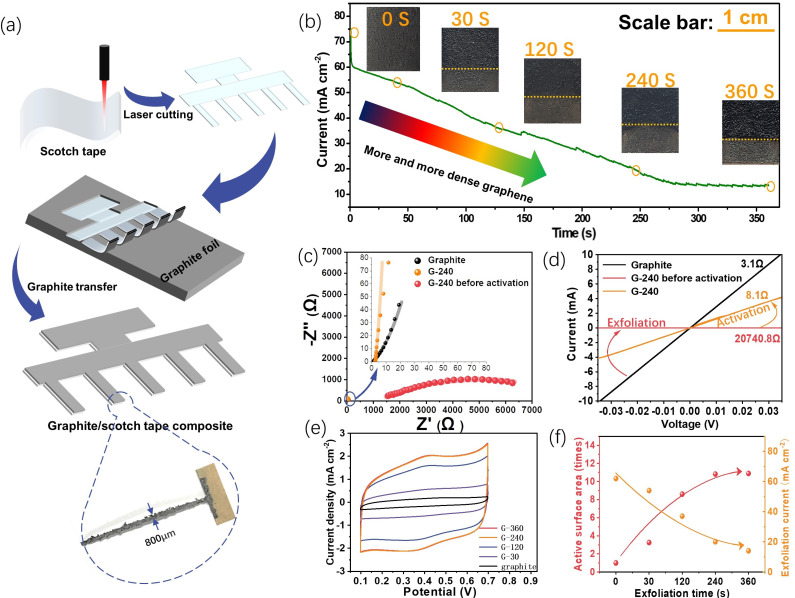
The preparation processes and some features of the planar graphene electrodes. (**a**) The illustration of the preparation process with an editable pattern; (**b**) the current measured during the preparation process (current density variation with time of exfoliation) and corresponding digital photos of the electrodes at certain exfoliation times; (**c**,**d**) the impedance spectra (**c**) and the I–V curves (**d**) of the graphite electrode and the graphene electrode with 240s exfoliation time, respectively, before and after activation; (**e**) the CV curves of the graphene electrodes prepared with different exfoliation times at 50 mV s^−1^ in 1M H_2_SO_4_ electrolyte (vs. Ag/AgCl 3 M); (**f**) the electroactive surface area and the exfoliation current for different exfoliation times obtained from (**b**,**e**), respectively.

**Figure 2 ijms-24-05599-f002:**
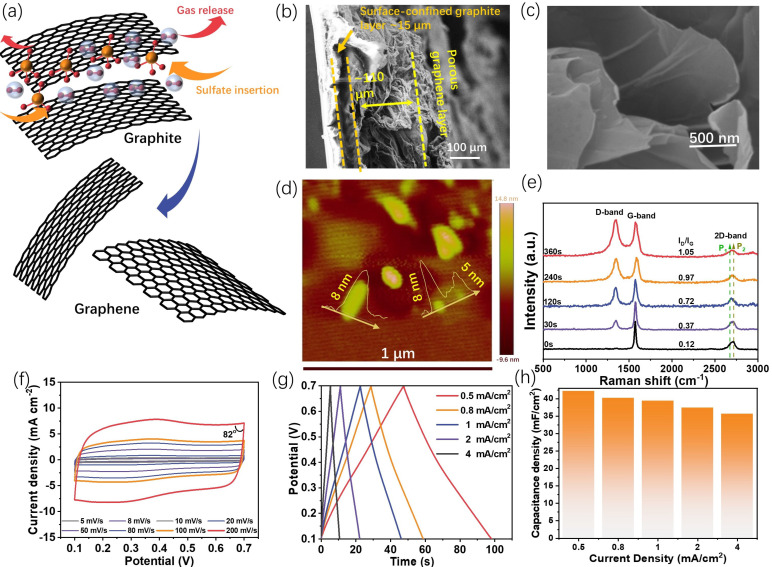
The detailed physiochemical properties of the prepared electrodes (especially G-240) and the energy storage abilities of G-240. (**a**) Illustration of the conversion from graphite to graphene during the electro-exfoliation process; (**b**,**c**) the respective cross-section (**b**) and high-resolution (**c**) SEM images of G-240; (**d**) the AFM image of the fakes from G-240; (**e**) the Raman spectra of the sets of graphene electrodes; (**f**) the CV curves of G-240 from the scan rates of 5 to 200 mV s^−1^ in the solution of 1M H_2_SO_4_ (vs. Ag/AgCl 3 M); (**g**,**h**) the charge–discharge curves and corresponding capacitance of G-240 at the current densities from 0.5 to 8 mA cm^−2^, respectively.

**Figure 3 ijms-24-05599-f003:**
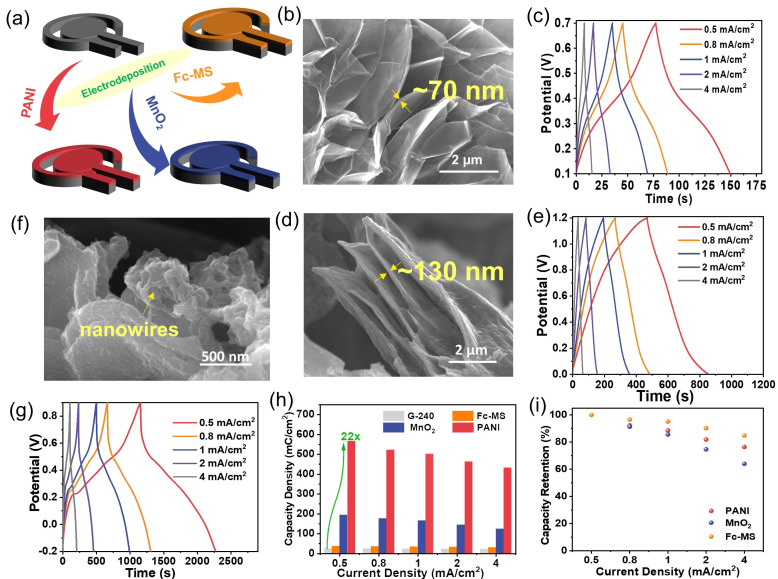
The preparation of the binary composites between G-240 and Fc-MS, MnO_2_, or PANI using electrodeposition methods. (**a**) The illustration of the composition between graphene and corresponding redox active materials; (**b**–**g**) the typical SEM images (**b**,**d**,**f**) and energy storage abilities (**c**,**e**,**g**) of G-240 functionalized with Fc-MS, MnO_2_, or PANI, respectively; (**h**,**i**) the calculated capacity density (**h**) and rate performance (**i**) of corresponding materials.

**Figure 4 ijms-24-05599-f004:**
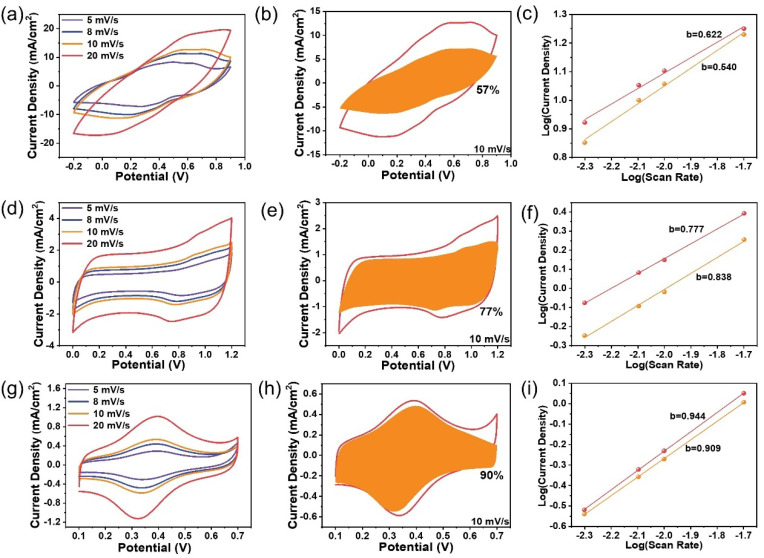
The kinetic analysis of the energy storage behaviors of Fc-MS, MnO_2_, and PANI: (**a**,**d**,**g**) the cyclic voltammetry of the three samples from 5 to 20 mV s^−1^; (**b**,**e**,**h**) the calculated surface-controlled current (the shaded area) from the respective CV curves (vs. Ag/AgCl 3 M); (**c**,**f**,**i**) the respective fitting slopes of the peak currents at different scan rates on a logarithmic scale.

**Figure 5 ijms-24-05599-f005:**
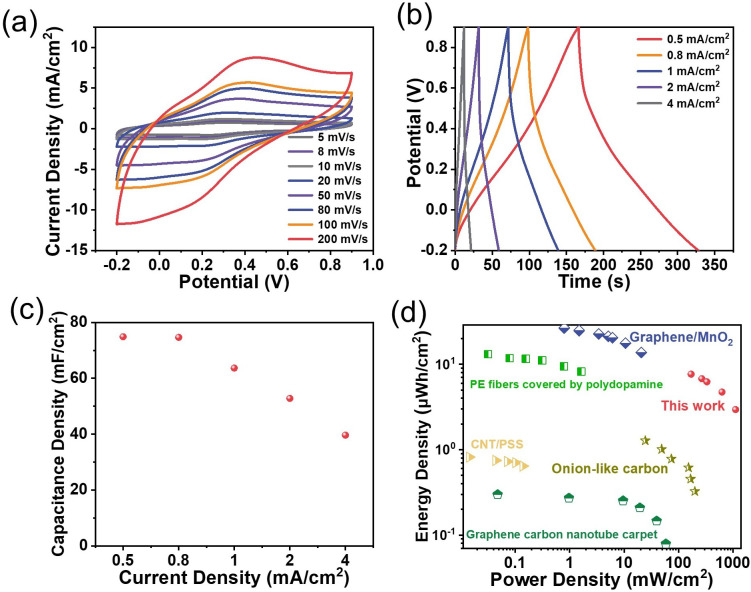
The energy storage performance of the PANI planar device in a symmetrical configuration: (**a**) the CV curves recorded at scan rates ranging from 5 to 200 mV s^−1^ in a potential window extending from −0.2 V to +0.9 V; (**b**) the charge–discharge curves at current densities of 0.5 to 4 mA cm^−2^; (**c**) the corresponding capacitance calculated from the charge–discharge curves; (**d**) the Ragone plot of this device and some reported benchmark devices: graphene carbon nanotube carpet [28]; CNT/PSS [29]; onion-like carbon [30]; PE fibers covered by polydopamine [31]; graphene/MnO_2_ [32].

## Data Availability

The data presented in this study are available on request from the corresponding author. The data are not publicly available due to the demand from our further research.

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
