# Peer review of "Laser Cutting Coupled with Electro-Exfoliation to Prepare Versatile Planar Graphene Electrodes for Energy Storage"

_ijms, 2023, doi:10.3390/ijms24065599_

Round 1
Reviewer 1 Report
The authors of paper which title is “Laser Cutting Coupled with Electro-exfoliation to Prepare Versatile Planar Graphene Electrodes for Energy Storage” evaluated the electro-exfoliation of planar graphene-based energy storage electrodes in different configuration. The study is interesting; however, authors need to consider the following list of revision before publishing the paper in energies:
1- The abstract part need to provide more quantitative results in terms of properties and electrochemical performance of the graphene-based electrode.
2- In terms of the exfoliation of graphene-based micro device, authors need to highlight the other electrochemical exfoliation method very briefly and then, highlight the advantage of current study. For that purpose, authors can use the following references:
https://doi.org/10.1039/C8NR08227H
https://doi.org/10.1016/j.jpowsour.2021.230701
3- The EIS data in Figure 1 (c) is not clear. Please enlarge the figure and also provide a equivalent circuit for the Nyquist curves with the value of each component in a separate table.
4- For the Raman data, Please plot the 2D bands part of Raman to find out the quality of exfoliated material.
5- In terms of electrochemical performance, please provide the CV data for very small scan rate such as 1 mV/s to check the redox peak much better. In terms of double layer type capacitor, please provide more in-details of the mechanism.
6- Please provide the loading of electrode for the pseudocapacitive materials fabrication.
7- In terms of nanowire formation on graphene using PANI, current SEM image is not clear. Please provide better quality SEM image of the morphology.
8- Please provide the EIS data after cycling for different systems.
9- Authors need to provide a comparison table for the graphene-based electrode for high performance capacitor applications and explain the state of the art data with current study.
Author Response
Please find our reply to reviewer 1 in the attached file.

Reviewer 2 Report
The manuscript reveals the facile top-down method to prepare graphene planer electrodes using laser cutting coupled with the electro-exfoliation method. The prepared electrodes show higher conductivity as well as a decent capacity density value. A further increment in the capacity has been observed by using a redox-active PANI-modified electrode. Higher pseudocapacitance is responsible for the increase in the higher energy density in the Ragone plot without compromising the power density. The study shows the practical applicability of work in the construction of flexible planer energy storage. The results in the manuscript are in support of demonstrating this practical application. However, there are still some concerns that should be addressed before publishing this article in this journal.
1. What is the effect on charge transfer resistance after electroexfoliation? Provide the fitted circuit for impedance including the extracted parameters.
2. What’s the behaviour of the electrode after the 360s?
3. Abstracted values of conductivities from figure 1d should be there in the manuscript.
4. In the voltammograms, mention the reference scale.
5. How did the author calculate the surface area? The procedure for that should be there in the experimental section.
6. Figure S3, assign all the peaks in XRD.
7. What is the stability of G240 after long-term electrochemical experiment? Provide any characterization to show the stability of that.
8. Line 276, correct this – 4a, 4d, and 4g.
9. Why are the CVs in figure 4a, 4d and 4g are different from S9 even at 20mV/s scan rate specially in case of PANI? Also, why are authors not seeing the electrochemical response of PANI in its potential window?
10. What is the effect on impedance in the cases of Fc-Ms, MnO2, and PANI?
11. Line 298-300, mention the figure number.
12. Compare the Ragone plot with G240.
13. Figure S6, why is the capacity retention greater than 100%?
Author Response
Please find our reply to reviewer 2 in the attached file.

Round 2
Reviewer 1 Report
The paper is ready to publish in the present form.